# Fibroblast Activation Protein Targeted Photodynamic Therapy Selectively Kills Activated Skin Fibroblasts from Systemic Sclerosis Patients and Prevents Tissue Contraction

**DOI:** 10.3390/ijms222312681

**Published:** 2021-11-24

**Authors:** Daphne N. Dorst, Arjan P. M. van Caam, Elly L. Vitters, Birgitte Walgreen, Monique M. A. Helsen, Christian Klein, Shreya Gudi, Tirza Wubs, Jyoti Kumari, Madelon C. Vonk, Peter M. van der Kraan, Marije I. Koenders

**Affiliations:** 1Department of Experimental Rheumatology, Radboud University Medical Centre, Daphne Dorst, P.O. Box 9101, 6500 HB Nijmegen, The Netherlands; Arjan.vanCaam@radboudumc.nl (A.P.M.v.C.); Elly.Vitters@radboudumc.nl (E.L.V.); birgitte.walgreen@radboudumc.nl (B.W.); monique.helsen@radboudumc.nl (M.M.A.H.); Shreya.Gudi@radboudumc.nl (S.G.); Peter.vanderKraan@radboudumc.nl (P.M.v.d.K.); marije.koenders@radboudumc.nl (M.I.K.); 2Roche Pharma Research and Early Development, Roche Innovation Center Zurich, 8952 Schlieren, Switzerland; christian.klein.ck1@roche.com; 3Institute for Molecules and Materials, Radboud University, 6525 AJ Nijmegen, The Netherlands; t.wubs@student.ru.nl (T.W.); Jyoti.Kumari@radboudumc.nl (J.K.); 4Department of Rheumatology, Radboud University Medical Center, P.O. Box 9101, 6500 HB Nijmegen, The Netherlands; madelon.vonk@radboudumc.nl

**Keywords:** fibroblasts, photodynamic therapy, systemic sclerosis, fibroblast activation protein

## Abstract

Systemic sclerosis (SSc) is a rare, severe, auto-immune disease characterized by inflammation, vasculopathy and fibrosis. Activated (myo)fibroblasts are crucial drivers of this fibrosis. By exploiting their expression of fibroblast activation protein (FAP) to perform targeted photodynamic therapy (tPDT), we can locoregionally deplete these pathogenic cells. In this study, we explored the use of FAP-tPDT in primary skin fibroblasts from SSc patients, both in 2D and 3D cultures. **Method:** The FAP targeting antibody 28H1 was conjugated with the photosensitizer IRDye700DX. Primary skin fibroblasts were obtained from lesional skin biopsies of SSc patients via spontaneous outgrowth and subsequently cultured on plastic or collagen type I. For 2D FAP-tPDT, cells were incubated in buffer with or without the antibody-photosensitizer construct, washed after 4 h and exposed to λ = 689 nm light. Cell viability was measured using CellTiter Glo^®®^. For 3D FAP-tPDT, cells were seeded in collagen plugs and underwent the same treatment procedure. Contraction of the plugs was followed over time to determine myofibroblast activity. **Results:** FAP-tPDT resulted in antibody-dose dependent cytotoxicity in primary skin fibroblasts upon light exposure. Cells not exposed to light or incubated with an irrelevant antibody-photosensitizer construct did not show this response. FAP-tPDT fully prevented contraction of collagen plugs seeded with primary SSc fibroblasts. Even incubation with a very low dose of antibody (0.4 nM) inhibited contraction in 2 out of 3 donors. **Conclusions:** Here we have shown, for the first time, the potential of FAP-tPDT for the treatment of fibrosis in SSc skin.

## 1. Introduction

Systemic sclerosis (SSc) is a rare, severe, systemic, auto-immune disease characterized by inflammation, vasculopathy and fibrosis of connective tissues. The disease can affect multiple organs, including the skin, gastro-intestinal tract and lungs, and ultimately results in loss of function in these organs. Such loss leads to high morbidity and a decreased life expectancy for patients. To date, treatment of (progressive) SSc consists mostly of immunomodulation with variable results [1]. In selected severe cases, patients may be eligible for autologous stem cell transplantation with good results, but most are not eligible [2], and an effective disease-modifying therapy suitable for all patients has not yet been identified.

Skin fibrosis is a hallmark of SSc, and is caused by presence and activity of myofibroblasts [3]. These cells are characterized by expression of contractile smooth muscle actin-myosin fibrils, and originate in the skin and organs of SSc patients due to the cytokine milieu present. Myofibroblasts produce large amounts of extracellular matrix (ECM) molecules, pro-fibrotic cytokines, and ECM strengthening enzymes, and are contractile. With this latter property, these cells can exert force on their surroundings and stiffen the extracellular matrix and consequently the skin. Until recently, it was thought that fibrosis was irreversible, but recent murine and human studies have shown that removal of myofibroblasts, e.g., by pro-apoptotic drugs, can normalize fibrotic tissues to a certain extent [4].

Targeting of myofibroblasts can be achieved via their expression of the prolyl endopeptidase fibroblast activation protein alpha (FAP). Expression of this membrane bound enzyme was found to be undetectable in most healthy adult tissues, but limited to activated fibroblasts including myofibroblasts [5]. Targeting of this molecule has been successfully used to eliminate pathological fibroblasts in preclinical disease models of arthritis and cancer [6,7]. However, systemic depletion of FAP-positive fibroblasts has been associated with the development of cachexia in animal studies due to inhibition of immune cell egression from bone marrow [8,9]. To overcome such potentially harmful effects of systemic FAP-positive cell depletion, we have recently developed and applied targeted photodynamic therapy (tPDT) to locoregionally eliminate FAP-positive cells in rheumatoid arthritis synovium. In tPDT a light sensitive molecule, i.e., the photosensitizer (PS), produces cytotoxic reactive oxygen species in response to light of a specific wavelength. By conjugating the PS to an antibody, for example one specific for FAP, this therapy can be used to selectively deplete FAP positive cells. A unique characteristic and major advantage of tPDT is that it allows for precision treatment to prevent unwanted systemic side effects: the treatment (the antibody-PS conjugate) can be administered systemically, but light exposure and thus activation of the drug can be limited to only carefully selected areas.

In this paper, we explored the use of FAP-tPDT on primary skin-derived (myo)fibroblasts of SSc patients, and show that tPDT can be used to successfully eliminate these cells in vitro in 2D and 3D culture. With demonstrating proof-of-concept of FAP-tPDT on SSc patient-derived fibroblasts, we have obtained the first functional in vitro evidence that supports the further development of this innovative technique for treatment of SSc.

## 2. Results

### 2.1. Fibroblasts from SSc Patient-Derived Skin Express FAP

In order to investigate if FAP was expressed by fibroblasts in the skin of SSc patients, both biopsies and fibroblasts derived from SSc patients’ lesional skin were stained for its presence. The biopsies showed clear presence of FAP staining in lesional skin as was demonstrated by immunohistochemistry (Figure 1A). In addition, we measured *FAP* gene expression in lesional skin of 25 SSc patients, and compared this to *FAP* expression in non-lesional skin (Figure 1B). This comparison showed that *FAP* is on average significantly more highly expressed in lesional skin, but of note, not in each individual patient per se. Whether a patient showed an increase in *FAP* expression in lesional skin versus unaffected skin was not correlated with age or modified Rodnan skin score, nor was it associated with a certain disease subtype or disease activity (see Supplemental Appendix A). Fibroblasts cultured from these lesional skin biopsies were also positive for FAP as visualized by immunofluorescence (Figure 1C), and expression of FAP expression was significantly enhanced by addition of 10 ng/mL TGFβ to the cell culture (Figure 1C,D).

### 2.2. Reduced Viability in SSc Skin Fibroblasts in Response to FAP-tPDT

Monolayer primary SSc skin fibroblasts were treated with FAP-tPDT at various antibody-PS concentrations. As controls, cells without antibody-PS, as well as cells that were not exposed to light, were included. The viability of cells incubated with antibody-PS construct and exposed to light was reduced in an antibody dose dependent manner. Already at a dose as low as 0.4 nM antibody-PS construct, cell viability was significantly reduced compared to non-exposed cells for all four donors (Figure 2). No dark toxicity was observed since cell viability is not affected at any antibody-PS dose in the plate not exposed to light (Figure 2A). Brightfield images of the cells after PDT showed loss of typical spindle-shape and nuclear blebbing (Figure 2B). Importantly, the effect of FAP-tPDT on cell viability is not mediated by aspecific binding of the PS to the cells as demonstrated by the lack of cytotoxicity when an aspecific antibody-PS construct (DP47GS-700DX) was used (Figure 2C).

### 2.3. Fibroblasts’ Culture Substrate Affects FAP-tPDT Efficiency

By culturing primary fibroblasts both on plastic and collagen, the influence of substrate on PDT efficiency was tested. Cells grown on collagen-coated wells were more resistant to FAP-tPDT than cells on plastic at low antibody-PS concentrations, indicating that substrate does influence the FAP-tPDT efficiency on SSc skin fibroblasts (Figure 3A). This effect is especially clear in cells that had always been cultured on collagen but were switched to plastic for this experiment (right panel in Figure 3B). These cells were significantly more sensitive to the cytotoxic effect of FAP-tPDT than those kept on collagen, as the LD_50_^′^s were 0.5 nM and 1 nM respectively.

### 2.4. Impaired Contraction of SSc Skin Fibroblasts after FAP-tPDT

To assess if FAP-tPDT is able to prevent the contraction of SSc fibroblasts, collagen plugs (Figure 4A) were seeded with primary SSc skin fibroblasts from three different donors and contraction of these plugs was monitored. In accordance with the monolayer results, FAP-tPDT decreased contractility of the fibroblasts since no contraction was observed in any of the donors at a dose of 10 or 2 nM antibody-PS (Figure 4B–D). Even when the cells were incubated with only 0.4 nM antibody-PS construct, the contraction was significantly impaired in two of the three donors (30.9% ± 2.8 versus 65.2% ± 6.1 and 41.5% ± 2.5 versus 78.6% ± 6.9 surface area remaining for unexposed versus exposed plugs from donor 2 and 3, respectively (*p* = 0.01 and <0.001 for donor 2 and 3, respectively)).

The levels of several cytokines (IL-1, -4, -6, -8, -10, IFNγ and TNFα) were determined in the supernatant of the plugs 5 h after FAP-tPDT. Only IL-6 and IL-8 measured above the detection limit. No changes in cytokine levels for either cytokine were detected when the plugs were not exposed to light, regardless of the presence or absence of antibody-photosensitizer (Figure 4E). In plugs that were exposed to light, but not to the antibody-photosensitizer construct, no difference in IL-6 or IL-8 production was observed, but in plugs incubated with a high dose of antibody-photosensitizer construct (2–10 nM) a decrease in cytokine production relative to the untreated controls was measured of approximately 30% and 75% for IL-6 and IL-8, respectively. Interestingly, plugs incubated with a lower dose of antibody-photosensitizer construct (0.08 or 0.4 nM) showed an increase in production of IL-6 and -8 relative to the untreated control of approximately 40% and 230% for IL-6 and IL-8, respectively (Figure 4E).

## 3. Discussion

Excessive fibrosis is a hallmark of systemic sclerosis. This fibrosis is caused in part by presence of (activated) myofibroblasts in tissues such as the skin [3]. In this paper we demonstrate that myofibroblasts obtained from lesional SSc skin can be targeted in vitro for elimination by photodynamic therapy using their FAP expression. With this method we were able to fully block myofibroblast contraction, an important pro-fibrotic process leading to skin hardening in SSc, in our 3D cell culture model.

Using histology and mRNA expression analysis, we were able to show that fibroblasts in lesional SSc skin express FAP. FAP expression in healthy human skin fibroblasts has not been reported by the human protein atlas (September 2021) [10], but has been reported in skin for activated fibroblasts associated with melanoma and during wound healing [5,11]. This fits with the concept that FAP is expressed in areas of active tissue remodeling, similarly to how inflammation can trigger FAP expression in synovial fibroblasts in rheumatoid arthritis [6]. Recent work has shown that in lesional skin of SSc patients, activated (cytotoxic) CD4+ T cells induce cell death of endothelial cells leading to tissue damage [12]. Such an environment could thus stimulate FAP expression. Unfortunately, no data has been reported on FAP expression in dermal fibroblasts in, e.g., dermatitis or systemic auto immune diseases with skin involvement such as systemic lupus erythematosus, psoriasis or arthritis psoriatica to support this idea. FAP positive fibroblasts have been identified in Dupuytren’s contracture, and the presence of myofibroblasts in this disease is associated with enhanced TNFα signaling [13,14].

Our data also shows that fibroblasts derived from lesional SSc skin express FAP in vitro. Fibroblasts cultured in vitro on plastic may differentiate into myofibroblasts and gain an activated phenotype due to elastic modulus of plastic providing a supernormal mechanical stimulus [15,16]. We made sure to include fibroblasts in our experiments that were never cultured on plastic, but that were cultured on collagen type 1 coated flasks instead. Cells cultured in this way still responded to FAP-tPDT, although higher conjugate concentrations were required to reach the same cytotoxic effect as for the cells cultured on plastic. It has been shown that myofibroblasts have mechanical memory in vitro, meaning that they preserve phenotypical traits fitting to a certain environment, e.g., stiff, even when placed in a new environment, e.g., soft [16]. Since FAP-tPDT was cytotoxic to SSc skin fibroblasts not only on stiff, but also on soft substrates, we hypothesize that the susceptibility of FAP expressing fibroblasts to FAP-tPDT we observe in vitro is a reflection of its in vivo sensitivity.

Elimination of myofibroblasts is expected to be a viable strategy to treat fibrosis [17]. For example, inhibiting dipeptydilpeptidase-4 (an enzyme structurally very similar to FAP) decreases bleomycin induced skin fibrosis in mice [18]. Furthermore, targeting of myofibroblasts in the same model using TLY012 (an engineered human TNF-related apoptosis-inducing ligand) reversed skin fibrosis [19]. In 3D skin fibroblast culture we were able to efficiently eliminate SSc fibroblasts using FAP targeted photodynamic therapy. Importantly, this led to decreased contraction of collagen hydrogels. Such contraction is a key contributor to tissue stiffness and fibrosis [20], because it creates a pro-fibrotic environment that mechanically stimulates the formation and survival of new myofibroblasts via mechanotransduction [4,21]. Contraction thus essentially creates a feed forward loop of increasing myofibroblast presence and activity. Of note, in SSc, these processes not only occur in skin, but also in other tissues such as lungs and the esophagus. Using FAP-tPDT, contraction could thus potentially be stopped and the fibrotic process reversed, not only in the skin, but also in other organs accessible to light.

Cytotoxicity in tPDT is determined by an interplay of many complicated factors including light penetration in tissue, available oxygen, perfusion of the tissue as well as photostability of the antibody-photosensitizer construct. To optimize these parameters, dosimetry may be used, but, despite considerable effort, there are no standardized procedures yet to study these complex phenomena [22]. An especially important factor when studying PDT in fibrotic skin is the oxygen tension. Especially the type II photoconversion reaction, in which molecular oxygen accepts the energy from the PS, which is dependent on oxygen tension [23]. All experiments described here were performed under standard, hyperoxic conditions, whilst it is known that the fibrotic skin of SSc patients can be hypoxic [24]. It is therefore possible that different oxygen concentrations may impact PDT efficiency.

Photodynamic therapy is a widely used technique to treat superficial non-melanoma skin cancers [25]. The PS currently approved for clinical use however are all untargeted and will therefore also result in damage to healthy cells in the area exposed to light. By using targeted PDT instead, this off-target toxicity can be limited. Furthermore, this technique will only result in cell death in those regions exposed to light, decreasing the chance of large systemic side effects. Of note, we did observe that sub-optimal treatment of skin fibroblasts in 3D with FAP-tPDT led to increased production of IL6 and IL8. These cytokines can activate and recruit local immune responses. The importance of this finding should be further investigated.

Treatment of fibrosis in SSc patients using FAP-tPDT does not have to be restricted only to the skin. Although the skin is the easiest organ to access with light, other options are also worth exploring in the future. As an example the esophagus may be exposed to light using an endoscopic system. This approach has already proven valuable in the treatment of Barret’s esophagus and early esophageal cancer [26].

## 4. Conclusions

In conclusion, we were able to show that FAP-tPDT is a powerful tool to eliminate myofibroblasts obtained from lesional SSc skin, thereby preventing contraction in our 3D skin fibroblast model. Therefore, FAP-tPDT may be a promising therapeutic avenue in the treatment of this devastating disease.

## 5. Materials and Methods

### 5.1. Collection and Processing of Patient-Derived Skin Biopsies

Primary skin fibroblasts were isolated from 4 mm skin biopsies from the lesional fibrotic parts of the forearm of SSc patients. The study protocol was approved by the local ethics committee (study number: NL57997.091.16). All patients provided written informed consent prior to the procedure. Biopsies were subsequently halved, and half was formalin fixed and paraffin embedded for histology, while the other half was placed in a 24 well plate in 2 mL DMEM Glutamax medium (Gibco, Waltham, MA, USA) supplied with 20% fetal calf serum, 100 U/mL penicillin, 100 mg/mL streptomycin and 100 mg/L pyruvate in standard culture conditions (5% CO_2_, 37 °C, 95% humidity) to allow for the outgrowth of primary fibroblasts, which occurs spontaneously within 14 days. Medium was partly refreshed once a week.

### 5.2. Cell Culture

Primary fibroblasts were cultured in DMEM Glutamax medium (Gibco) supplied with 10% fetal calf serum, 100 U/mL penicillin, 100 mg/mL streptomycin and 100 mg/L pyruvate in standard culture conditions. Medium was partially refreshed (50%) every third day. Cells were used in experiments starting from passage 5. Cells from 3 donors were cultured on plastic, whereas cells from 1 donor were cultured either on plastic or on collagen (PureCol collagen type I (Advanced BioMatrix, Carlsbad, CA, USA)). For stimulation, TGFβ1 (Biolegend, San Diego, CA, USA) was used and this stimulation was also partially refreshed (50%) every third day.

### 5.3. Antibody Conjugation

The monoclonal, FAP-targeting, antibody 28H1 was conjugated to the photosensitizer IRDye700DX (LiCor) at a molar excess of 5 PS molecules per antibody (Ab-PS), using a N-hydroxysuccinimidyl ester, as previously described [27]. Briefly, the antibody and PS were incubated in 10% *w/v* 1 M NaHCO_3_ (pH 8.5) at room temperature (RT) for 1 h. Subsequently, the chelator ITC-DTPA was conjugated to the Ab-PS construct at a 10-fold molar excess and incubated in 10% *w/v* 1 M NaHCO_3_ (pH 9.5) at RT for 1 h to allow for radiolabeling with indium-111. IRDye700DX and ITC-DTPA that were not conjugated to the antibody, were removed through dialysis with a Slide-A-Lyzer Dialysis Cassette (20,000 MWCO, Thermofisher, Waltham, MA, USA) in phosphate buffered saline (PBS) with 0.5% *w/v* Chelex (Bio-Rad, Hercules, CA, USA). The final substitution ratio (SR) of the Ab-PS construct was approximately 2.5 PS molecules per Ab. Unless specified the compounds used were not radiolabeled.

### 5.4. In Vitro 2D FAP-tPDT

For in vitro tPDT, 2 × 10^5^ primary fibroblasts from three donors were seeded in 96-wells plates. After overnight culture to ensure cell adherence, the cells were incubated with DTPA-28H1-700DX in a range of 0–30 nM in binding buffer (DMEM supplemented with 0.5% bovine serum albumin (BSA, Sigma-Aldrich, St. Louis, MI, USA) (BB)) for 4 h at 37 °C. Control cells were incubated with BB alone or with an irrelevant IgG1 antibody with the same PGLALA mutation as 28H1 (DP47GS) also conjugated with IRDye700DX, as indicated in the figures. After incubation, the cells were washed and exposed to a light dose of 50 J/cm^2^ at 280 mW/cm^2^ radiance of 690 nm light from a LED light source (LEDfactory, Leeuwarden, The Netherlands) [28]. After light exposure, the cells were incubated overnight at 37 °C with 5% CO_2_. The medium was subsequently removed and CellTiter-Glo (Promega, Madison, WI, USA) was used to measure ATP levels. Cell viability was expressed as the mean percentage of luminescence compared to the non-exposed BB control group.

Cell viability was also measured using the MTT assay (Invitrogen, Waltham, MA, USA). Cells were exposed to FAP-tPDT and subsequently incubated with 12 mM 3-(4,5-dimethylthiazol-2-yl)-2,5-diphenyltetrazolium bromide for 4 h at 37 °C with 5% CO_2_. The formazan crystals were dissolved using dimethyl sulfoxide and absorbance was subsequently measured at 570 nm, background absorbance at 650 nm was deducted from the signal and this was normalized to the control not incubated with antibody and not exposed to light.

### 5.5. FAP-tPDT on Different Substrates

Substrate stiffness is an important determinant of fibroblast activation. To investigate the effect of this on FAP-tPDT efficiency, cell viability of cells plated on collagen coated wells was compared to that of cells seeded on plastic. PureCol collagen type I (Advanced BioMatrix, Carlsbad, CA, USA) was used to coat 48-well plates as per the manufacturer’s instructions. The liquid collagen was syringe filtered using a 0.2 µ filter (VWR International, Radnor, PA, USA). Cells at a density of 2.5 × 10^5^ cells/well were seeded in culture medium on top of the collagen or plastic matrix and allowed to adhere for one day at 37 °C with 5% CO_2_. The next day, the cells were washed and BB was added with 28H1-700DX or plain BB as control. After 4 h of incubation the cells were washed once and BB was added. The cells were subsequently exposed to 52 J/cm^2^ of 689 nm light at a fluence of 290 mW/cm^2^. As a control, one plate for each condition was not exposed. Cell viability was measured the next day using CellTiterGlo (Promega, Madison, WI, USA). The percentage of viable cells was calculated based upon the luminescence measured in the cells not incubated with antibody and not exposed to light.

### 5.6. In Vitro 3D FAP-tPDT on Collagen Plugs

To make 3D collagen hydrogels, the primary fibroblasts were detached using trypsin and brought to a concentration of 2 × 10^6^ cells/mL. Next, 20 μL Minimal Essential Medium (Sigma-Aldrich, Saint Louis, CA, USA), 10 μL sodium bicarbonate (Gibco, Waltham, MA, USA), 150 μL soluble collagen (PureCol, type 1 collagen) and 90 μL cell suspension were added per plug in a separate tube in respective order. The suspension was carefully homogenized and 250 μL of the suspension was added per well of a 48-well plate. The plugs were incubated for one hour under standard culture conditions to solidify them, after which, 750 μL of colorless DMEM medium (Gibco) supplemented with penicillin, streptomycin and pyruvate (Invitrogen, Waltham, MA, USA) was added. The next day, 10 uL of medium containing antibody-PS at concentrations ranging from 0 to 10 nM (final concentration) was added. After overnight incubation, the plugs were washed once with medium and subsequently exposed to 52 J/cm^2^ of λ = 689 nm light at a fluence of 290 mW/cm^2^. As a control, one plate for each condition was not exposed. After the exposure, the plates were put back into the stove at 37 °C with 5% CO_2_ for 1 h. Using a blunt 27 gauge needle, the plugs were detached from the rim of the well plate and allowed to contract for 5 h. At 1, 2, 3, 4, 5 h post-detachment the contraction was measured by scanning the plates on a standard office flat-bed scanner. To determine the area, the images were analyzed using Fiji ImageJ.

### 5.7. Immunohistochemistry

The formalin fixed and paraffin embedded skin biopsies were cut in 5 µm sections and stained for FAP (LN031634, LabNed, Antwerp, Belgium, 1/100, PBS/1%BSA). The slides were deparaffinized by xylene wash and rehydrated using ethanol. Antigen was retrieved using 10 mM citrate (pH 6.0). The peroxidase activity was blocked by incubating with 3% H_2_O_2_ in PBS. Non-specific binding was blocked through preincubation with 20% NGS for 30 min. Next the slides were incubated with the primary antibody for 1 h at room temperature. Slides were incubated with the secondary biotinylated goat anti-rabbit antibody (VECTASTAIN, Thermo-Fisher, Waltham, MA, USA, 1/200 in PBS/1% BSA) followed by labelling with the avidin-biotin complex (VECTASTAIN, Thermo-Fisher, Waltham, MA, USA, 1/100). The antibody complex was visualized using diaminobenzene (bright DAB, Immunologic). All slides were counterstained with hematoxylin and mounted with a cover slip (Permount, Thermo-Fisher, Waltham, MA, USA).

### 5.8. Immunofluorescence

For the immunofluorescent staining, primary skin fibroblast cells were seeded at a density of 1 × 10^3^ cells on an ibiTreat µ-Slide angiogenesis dish (Ibidi, Munich, Germany) and cultured until day 5. The plates were precoated with Collagen (Collagen R solution 0.2%; 1:20 in PBS). Cells were subsequently fixed using 10% formalin for 10 min and permeabilized using 0.5% (*v*/*v*) Triton X-100 in PBS for 20 min. After washing, the cells were incubated in blocking buffer (2% *w/v* BSA, 2% normal donkey serum, 0.025% *v/v* Triton-X, 0.05% Tween-20) with 100 mM glycine for 30 min. Next, the primary anti-FAP antibody (LN031634, LabNed, Antwerp, Belgium, 1/250 in blocking buffer) was added and the cells were incubated overnight at 4 °C. The next day, after washing, the secondary antibody (goat anti-rabbit-Alexa fluor 488 (1:200, Invitrogen, Waltham, MA, USA)) was incubated for 1 h at room temperature. Cells were subsequently washed using PBS with 0.05% *v/v* Tween-20 and rinsed in PBS. DAPI (1/100, Sigma) was added to stain the nuclei. After a quick rinse in PBS, the fluorescent images were acquired using a SP8x AOBS-WLL confocal microscope (Leica Microsystems, Mannheim, Germany).

### 5.9. RNA Isolation and Quantitative Real-Time PCR

Skin biopsies were collected from 25 SSc patients after informed consent study number: NL57997.091.16. RNA was extracted using RNeasy Fibrous Tissue Mini Kit (Qiagen, Hilden, Germany) according to the manufacturer’s protocol. Next, RNA concentration was measured using a Nanodrop photospectrometer (Thermo Scientific, Waltham, MA, USA), and subsequently, 1 μg of RNA was converted into cDNA in a single step reverse transcriptase PCR using oligo dT primer and M-MLV Reverse transcriptase (Life Technologies, Carlsbad, CA, USA). In this cDNA, *FAP* gene expression was measured using 0.2 µM of validated cDNA-specific primers (see Table 1) (Biolegio, Nijmegen, the Netherlands) in a quantitative real time polymerase chain reaction (qPCR) using SYBR green master mix (Applied Biosystems, Waltham, MA, USA). Relative gene expression (−ΔCt) was calculated using four reference genes: *GAPDH*, *HPRT*, *TBP*, and *RPS27A*.

### 5.10. Cytokine Measurements

Human cytokines and chemokines in culture supernatant were measured by Luminex using BioPlex kits according to manufacturer’s instructions, and analyzed using BioPlex Manager 4 software (Bio-Rad Laboratories, Hercules, CA, USA).

### 5.11. Statistics

All quantitative data are expressed as a mean of multiple repeats ± SD. For every analysis, data was checked for normality using the Shapiro–Wilk test. Student’s t-test or one-way analysis of variance (ANOVA) with Tukey multiple comparison post-test was used to determine significance. If needed, *p* values were corrected for multiple testing using Bonferroni correction. The statistical analyses were performed using GraphPad Prism software (version 5.0, San Diego, CA, USA)

## Figures and Tables

**Figure 1 ijms-22-12681-f001:**
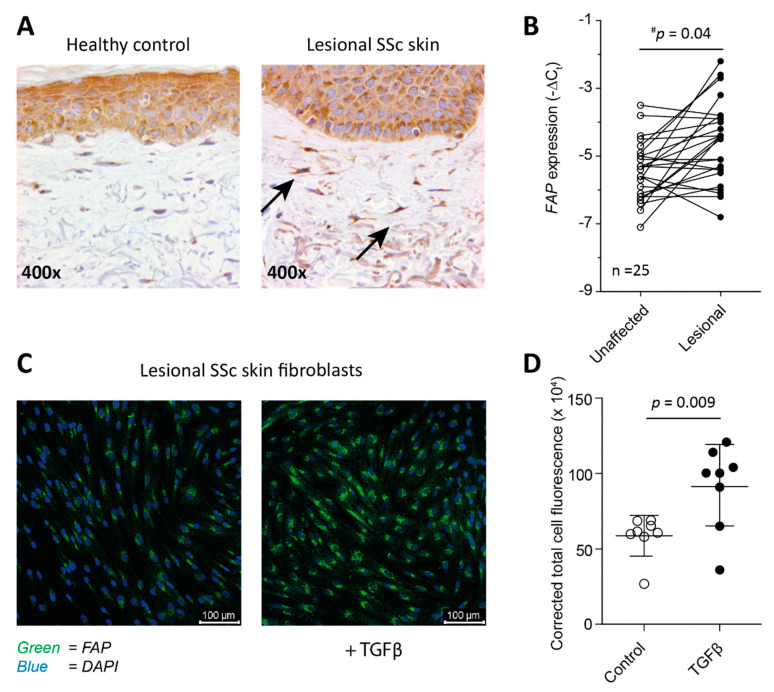
(Myo)Fibroblasts in lesional SSc skin express FAP. Skin biopsies stained for FAP using IHC show presence of FAP positive fibroblasts (black arrows) in lesional SSc skin (**A**). On average, *FAP* gene expression is significantly increased in lesional SSc skin compared to unaffected skin of the same patients as determined by a one-sided paired Students’ T-test. The reported ^#^
*p* value is a corrected *p* value, the uncorrected *p* value is 0.0013 (**B**). Primary fibroblasts obtained from lesional SSc skin express FAP in vitro as detected by immuno-fluorescence microscopy (**C**), and FAP expression is significantly increased by stimulation with 10 ng/mL TGFβ for 5 days (**D**).

**Figure 2 ijms-22-12681-f002:**
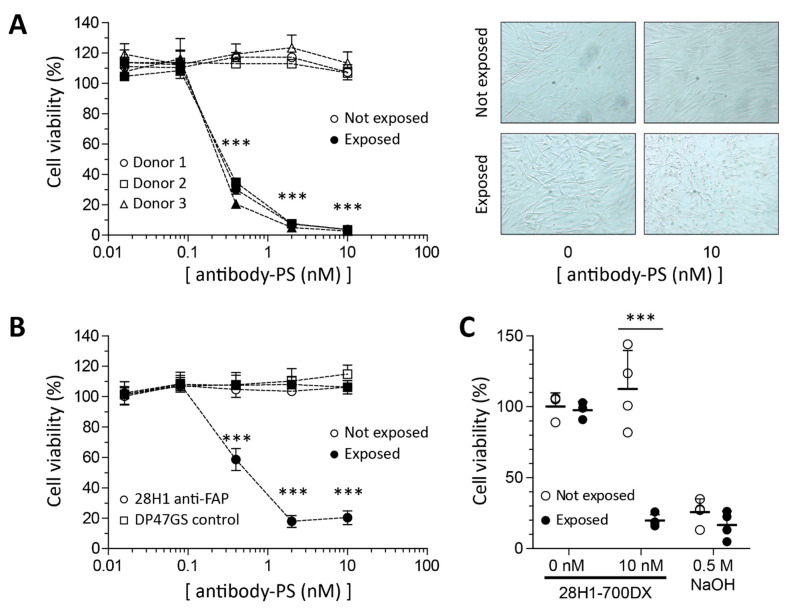
Application of FAP-tPDT in monolayer culture of primary lesional SSc skin fibroblasts. In three fibroblast cell strains, a dose range of PS-FAP antibody construct was applied and cell viability was measured using CTG after exposure to light and compared to cells not exposed to light. The PS-FAP antibody construct dose-dependently killed fibroblasts with a LD_50_ of approximately 0.25 nM but only in cells exposed to light (**A**). Application of a PS-non-targeting antibody (DP47GS) construct had no effect on fibroblast viability (**B**). (*n* = 4 replicates per donor). Cell viability decreases after FAP-tPDT as measured by the MTT assay. Cells were incubated with 0 or 10 nM 28H1-700DX or 0.5 M NaOH. No cell death was observed in the cells not exposed to light or not incubated with the antibody. Results are depicted as mean with SD (*n* = 4 or *n* = 3 for cells not incubated with the antibody (0 nM)) (**C**). *** = *p* < 0.001.

**Figure 3 ijms-22-12681-f003:**
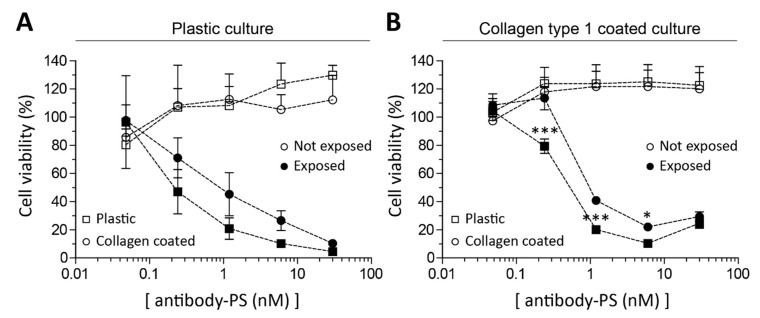
Application of FAP-tPDT in monolayer culture of primary lesional SSc skin fibroblasts cultured on different substrates. Primary skin fibroblasts were isolated and cultured on either plastic (**A**) or collagen type 1 (**B**) and subsequently seeded on plastic or collagen type 1 to determine the dose response of PS-FAP antibody construct in these conditions after exposure to light. For plastic cultured cells the LD_50_ of the tPDT construct was 0.25 nM and 1 nM for cells seeded on plastic and collagen respectively. For collagen cultured cells, the LD_50_ of the tPDT construct was 0.5 nM and 1 nM respectively. (*n* = 4 replicates per donor). * = *p* < 0.05, *** = *p* < 0.001.

**Figure 4 ijms-22-12681-f004:**
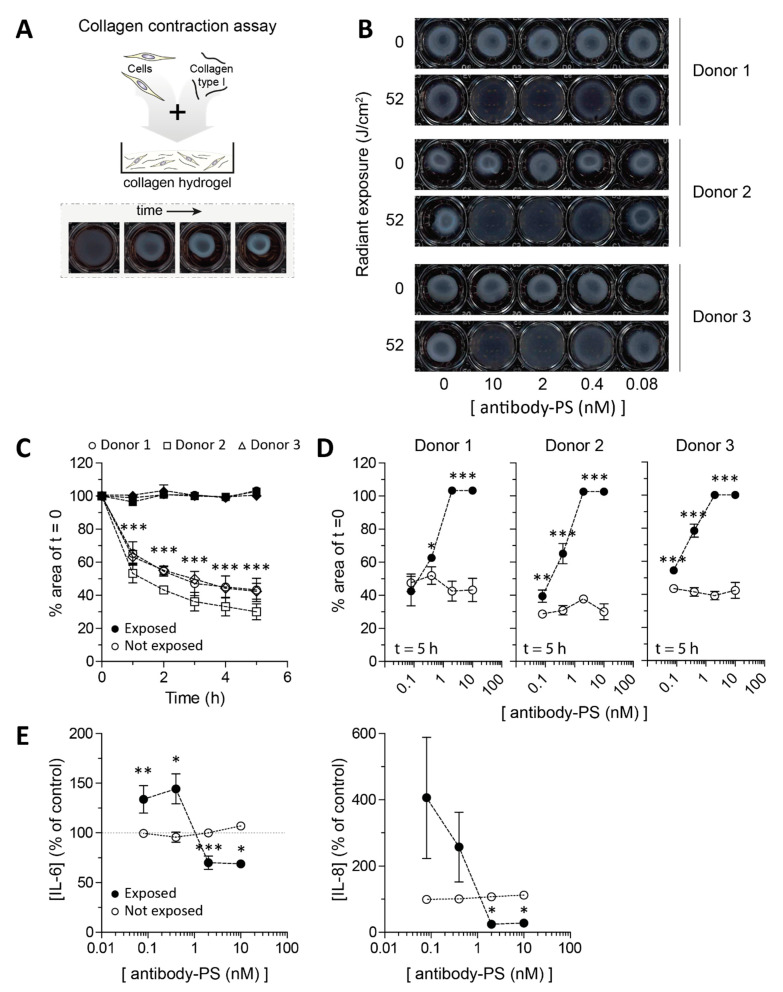
Application of FAP-tPDT in 3D culture of primary lesional SSc skin fibroblasts cultured in collagen type 1 hydrogels. Primary skin fibroblasts of 3 different donors were seeded in collagen type 1 hydrogels (**A**) and exposed to a dose response of tPDT after 48 h. Subsequently contraction was visualized after 5 h (**B**) and quantified over time for a dose of 10 nM PS-antibody construct. In hydrogels not exposed to light, contraction occurs but this is fully blocked by tPDT (**C**). Dose response curves for each individual donor after 5 h are depicted in (**D**) (all *n =* 4 replicates per donor). IL6 and IL8 levels were measured in the supernatant of the hydrogels 5 h after tPDT procedure (**E**). (Pooled data from duplicates of all 3 donors). * = *p* < 0.05, ** = *p* < 0.01, *** = *p* < 0.001.

**Table 1 ijms-22-12681-t001:** Primer sequences used in this study. Reference genes are marked with an asterisk (*).

Gene	Forward Primer 5′-->3′	Reverse Primer 5′-->3′
*GAPDH **	ATCTTCTTTTGCGTCGCCAG	TTCCCCATGGTGTCTGAGC
*HPRT **	CCTGGCGTCGTGATTAGTGA	TCTCGAGCAAGACGTTCAGT
*TBP **	GCTTCGGAGAGTTCTGGGATTG	GCAGCAAACCGCTTGGGATTA
*RPS27A **	TGGCTGTCCTGAAATATTATAAGGT	CCCCAGCACCACATTCATCA
*FAP*	GCTTTGAAAAATATCCAGCTGCC	ACCACCATACACTTGAATTAGCA

## Data Availability

All data generated or analyzed during this study are in the manuscript and in the Appendix A.

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
