# Peer review of "Fibroblast Activation Protein Targeted Photodynamic Therapy Selectively Kills Activated Skin Fibroblasts from Systemic Sclerosis Patients and Prevents Tissue Contraction"

_ijms, 2021, doi:10.3390/ijms222312681_

Round 1

Reviewer 1 Report

The authors presented the results of preliminary studies on the use of targeted photodynamic therapy against myofibroblasts in systemic sclerosis.

The presented results are very promising and constitute the basis for further research into usage of targeted photodynamic therapy, not only in systemic sclerosis, but also in other diseases associated with excessive skin fibrosis.

However, there are some issues that need clarification:

  • The title may be a bit confusing, it should clearly show the nature of this "in vitro" research
  • Layout should be adapted to the requirements of the journal
  • Authors stated that "This comparison showed that FAP is on average significantly higher expressed in lesional skin, but of note, not in each individual patient per se." Analyzing Figure 1 B - in some patients higher FAB expression was found in unaffected skin. What did it depend on? Disease activity, disease duration, type of systemic sclerosis (limited vs diffuse)? Perhaps it would be worth adding a table (to the text or as a supplementary data) containing the clinical characteristics of the patients. These may be important data for the future use of this method in therapy.
  • In the context of the extra-cutaneous application of therapy, how deep can light penetrate? The use in pulmonary fibrosis may seem overly optimistic.

Author Response

Please find our cover letter and point-to-point response to the reviewers in the attachment

Reviewer 2 Report

In this manuscript, Dorst, van Caam and colleagues have devised an elegant approach to target fibroblasts. Using anti-FAP in conjunction with tPDT to delete fibroblasts by conjugating anti-FAP antibody to PS. The advantage of the approach is that it avoids the systemic toxicity noted with systemic depletion of myofibroblasts by exerting the effect locally/regionally. The approach is rigorous and shows anti-FAP treated cells with and without light exposure as well as cells treated with non-targeting antibody conjugated to PS with and without light exposure. The ‘n’ is small but acceptable for a proof of concept manuscript. The approach is innovative and would be greatly strengthen by the inclusion of parallel data from experiments done with healthy donor fibroblasts with and without TGFb treatment.

Major comments:

One of the limitations of the study is that it is only a demonstration of the invitro effect of anti-FAP-PS. Ultimately, in vivo efficacy data with no toxicity is essential to obtain.

Figure 1C should include immunofluorescent detection of FAP in healthy donor fibroblasts for comparison (with and without TGFb).

Figure 1, since the authors state that the fibroblasts are from only 3 donors, the figure legend should specify that and explain how many replicates of each donor are shown in the graphical data. The same applies to other figures.

The data could be strengthened by a comparison of FAP levels and the response of healthy donor fibroblasts (with and without TGFb) to anti-FAP-PS/light exposure.

It appears from Figure 2 as if exposure to the light with no anti-FAP antibody (image labeled as 0) shows some effect on fibroblasts.

Also, in Figure 2, is there a way to measure the percentage of cells responding to the conjugate/light combination and undergoing apoptosis?

Since FAP expression is variable in lesional skin of different patients, how do the authors envision standardizing this as a therapy for patients, especially those who have fewer FAP+ fibroblasts?

Overall, it is important to confirm loss of cell viability via a second assay.

Minor comments:

Fibroblasts are treated with TGFb for 5 days. Is the effect of TGFb sustained for that long or is the fibrotic phenotype of the cells beginning to reverse?

The authors may want to delete ‘n=4’ from the abstract and reserve the information for the methods and figure legends.

Please include the duration of light exposure in the methods.

Author Response

(The authors gave the same response as above.)
